# Escape mutations circumvent a tradeoff between resistance to a beta-lactam and resistance to a beta-lactamase inhibitor

Dor Russ [1], Fabian Glaser[2], Einat Shaer Tamar[1], Idan Yelin [1], Michael Baym [3], Eric D. Kelsic [4], Claudia Zampaloni[5], Andreas Haldimann[5] & Roy Kishony [1,6✉]

Beta-lactamase inhibitors are increasingly used to counteract antibiotic resistance mediated by beta-lactamase enzymes. These inhibitors compete with the beta-lactam antibiotic for the same binding site on the beta-lactamase, thus generating an evolutionary tradeoff: mutations that increase the enzyme's beta-lactamase activity tend to increase also its susceptibility to the inhibitor. Here, we investigate how common and accessible are mutants that escape this adaptive tradeoff. Screening a deep mutant library of the $bla_{ampC}$ beta-lactamase gene of *Escherichia coli*, we identified mutations that allow growth at beta-lactam concentrations far exceeding those inhibiting growth of the wildtype strain, even in the presence of the enzyme inhibitor (avibactam). These escape mutations are rare and drug-specific, and some combinations of avibactam with beta-lactam drugs appear to prevent such escape phenotypes. Our results, showing differential adaptive potential of $bla_{ampC}$ to combinations of avibactam and different beta-lactam antibiotics, suggest that it may be possible to identify treatments that are more resilient to evolution of resistance.

[1] Faculty of Biology, Technion-Israel Institute of Technology, Haifa, Israel. [2] Lorry I. Lokey Interdisciplinary Center for Life Sciences and Engineering, Technion-Israel Institute of Technology, Haifa, Israel. [3] Department of Biomedical Informatics, Harvard Medical School, Boston, MA, USA. [4] Department of Genetics, Harvard Medical School, Boston, MA, USA. [5] Roche Pharma Research and Early Development, Immunology, Infectious Diseases, and Ophthalmology, Roche Innovation Center Basel, F. Hoffmann-La Roche Ltd, Grenzacherstrasse 124, 4070 Basel, Switzerland. [6] Faculty of Computer Science, Technion-Israel Institute of Technology, Haifa, Israel. ✉email: rkishony@technion.ac.il

Resistance to beta-lactam antibiotics, mediated primarily through beta-lactamases, is growing as a major threat to public health. Beta-lactams are commonly used for the treatment of a range of bacterial pathogens, constituting the most widely prescribed antibiotic class[1,2]. These antibiotics are characterized by a core of a beta-lactam ring and are classified by their moieties into four major families: penicillins, cephalosporins, carbapenems, and monobactams[3]. The irreversible binding of these drugs to the peptidoglycan cross-linking enzymes (Penicillin Binding Proteins, PBPs) leads to cell death and lysis. Resistance to beta-lactams has become widespread in recent years, mainly through drug degradation by beta-lactamases, most notably by the serine beta-lactamases of classes A and C[4,5]. Hydrolyzing the core beta-lactam ring, these enzymes decrease the effective drug concentration, thereby conferring increased resistance[6–8]. The effectiveness of these enzymes is often further improved either by mutations that increase their expression or by intragenic structural mutations that enhance their efficacy and specificity[9–14]. To overcome beta-lactamase mediated antibiotic resistance, beta-lactam antibiotic treatments are often supplemented by a beta-lactamase inhibitor[15–17]. These inhibitors compete with the drug for the same binding site, yet being resilient to degradation they block enzyme activity thereby restoring antibiotic efficacy. While class A beta-lactamase inhibitors, such as clavulanate or tazobactam, are commonly used, efficient inhibitors of the widespread class C beta-lactamases have only recently been introduced.

Class C beta-lactamases confer resistance to broad-spectrum cephalosporins, penicillins, and monobactams[18,19]. Like other beta-lactamases, they are highly transferable and widespread among Gram-negative bacteria, especially in clinical isolates[20,21]. Class C beta-lactamases catalyze the hydrolysis of the beta-lactam ring by a conserved catalytic Serine (Ser 64 in *Escherichia coli* $bla_{ampC}$) and an activating conserved Tyrosine (Tyr 150)[19,22–24]. A major obstacle to effective treatment of pathogens that carry class C beta-lactamases is the insensitivity of the enzyme to classical beta-lactamase inhibitors; the different structure of the pocket of class C enzymes compared to class A prevents the classical inhibitors from binding to its pocket[25,26]. Avibactam is a novel non-beta-lactam molecule that inhibits the activity of class C as well as of other beta-lactamases[27–29]. It is resilient to class C beta-lactamase hydrolysis and inhibits their activity by binding the catalytic Serine covalently yet reversibly[28]. Avibactam can substantially decrease the resistance level of class C beta-lactamase-carrying bacteria to beta-lactam drugs[30]. Indeed, avibactam was recently approved for clinical use in combination with the cephalosporin ceftazidime and a combination of avibactam with the monobactam aztreonam is currently in clinical trials[31,32]. It is therefore important to understand the potential for evolutionary adaptation of the enzyme to combinations of avibactam with different beta-lactams.

Resistance to beta-lactamase inhibitors is typically associated with increased susceptibility to beta-lactam drugs, representing inherent constraints on adaptive mutations. While mutations that increase beta-lactamase expression can increase resistance to the drug alone and in combination with the inhibitor, resistance to the drug-inhibitor treatment via structural intragenic mutations is inherently constrained: due to the structural similarity between the drug and the inhibitor, mutations that increase drug degradation also tend to increase the affinity of the enzyme to the inhibitor[7,33–38]. Of course, this functional tradeoff between resistance to the drug and the inhibitor does not completely preclude mutations that increase resistance to the combination[39–42]. Indeed, even without escaping this tradeoff, a mutation providing strong resistance to one compound, even at the cost of mild sensitivity to the other, can provide an overall increased resistance

to the combination[34,41,42]. Less is known though about mutations that escape this tradeoff, allowing the mutant to survive, even in the presence of the inhibitor, at drug concentrations that kill the wild-type in the absence of the inhibitor (native inhibitory concentration). In particular, it is unclear how prevalent are such escape mutations, how specific they are to different beta-lactams and whether they might be accessible via a single amino-acid substitution.

Here, focusing on *E. coli* Bla$_{ampC}$ enzyme as a model for class C beta-lactamases, we constructed a systematic single amino-acid substitution mutant library and identified resistance and escape mutations to avibactam paired with different beta-lactam drugs. Measuring growth of the mutants across gradients of different drugs, with and without avibactam, we found that escape mutations are accessible but only for some drugs and not others. These escape mutations allow bacteria to grow even at the presence of avibactam at the high drug concentrations required to kill the wild-type without avibactam. Sequencing the selected mutant library, we identified these escape mutations and found that they are rare and drug-specific.

## Results

**Deep mutant library of $bla_{ampC}$.** To systematically study the tradeoffs between mutations that confer resistance to beta-lactam and beta-lactamase inhibitors, we constructed a library of mutants inside the beta-lactamase encoding gene, $bla_{ampC}$. First, we identified in silico 44 residues at and in the near vicinity of the active pocket of the Bla$_{ampC}$ enzyme (Methods; Supplementary Fig. 1). Next, starting with an *E. coli* parental strain expressing the chromosomal $bla_{ampC}$, we systematically mutated these intragenic positions using Multiplexed Automated Genome Engineering (MAGE, Fig. 1a; Methods)[43–45]. Our design guaranteed that each mutated codon differed from the wild-type codon by 2–3 nucleotides, thereby much reducing misidentifications caused by sequencing errors (Methods, Supplementary Data 1; this 2-mutation constraint still allows access to the vast majority of possible amino-acid changes: 846 out of the potential $44 \times (19$ amino acids + stop$) = 880$ single substitution mutants, 96%). The mutant library contained 93.4% of these designed single amino-acid substitutions, with mutant frequencies distributed with an average and standard deviation of $5.3 \times 10^{-5} \pm 4.2 \times 10^{-5}$ (as identified by amplicon sequencing using Illumina's MiSeq, Methods, Supplementary Fig. 2).

**Escape phenotype is observed by growth measurements.** To characterize the level of resistance that mutations in $bla_{ampC}$ can confer, we pooled all the MAGE mutants and selected the pooled library on gradients of five different antibiotics, representing all major beta-lactam families: the penicillins piperacillin (PIP) and ampicillin (AMP); the monobactam aztreonam (ATM); the cephalosporin cefepime (FEP); and the carbapenem meropenem (MER). Each drug was applied with or without the beta-lactamase inhibitor avibactam (AVI). Culture density was monitored over time for the mutant library as well as the parental strain (WT, expressing unmutated $bla_{ampC}$). As drug concentration increased, the cultures took longer to reach detectable OD, yet grew at similar rates (Fig. 2a). To characterize these time delays we measured $t_{th}$, the time in which culture density crossed a set threshold, $OD_{th}$. Next, for each drug concentration, we represented the measured delay by an estimated initial density of resistant mutants $OD_{Res}^0 = OD_{th} \cdot e^{-g \cdot t_{th}}$, where $g$ is bacterial growth rate (Fig. 2a; Supplementary Fig. 3). Plotting $OD_{Res}^0$ against drug concentration, we defined the critical drug concentration $\eta$ at which the expected initial density of the mutants drops below an OD value corresponding to the expected number

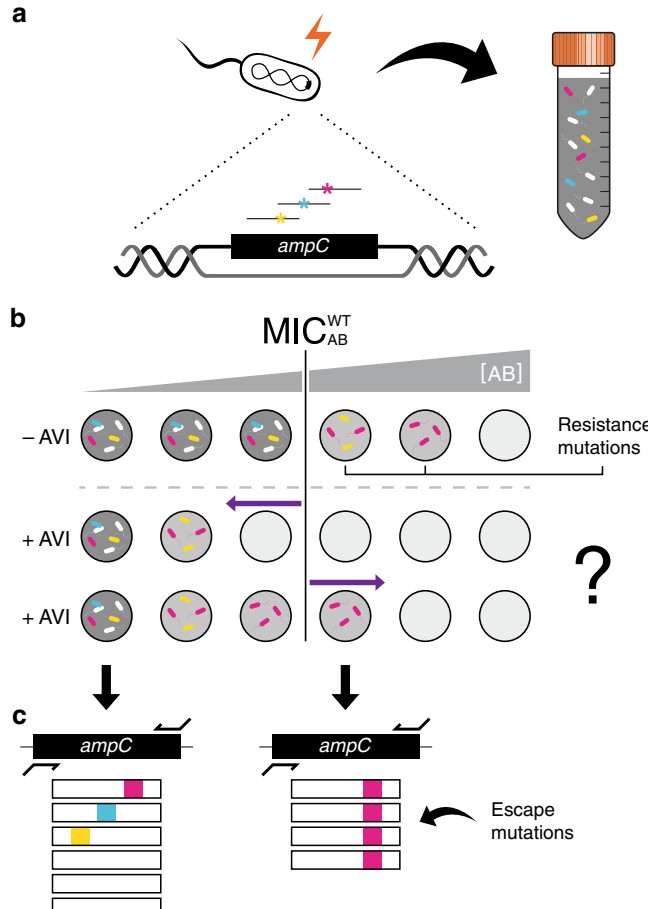

**Fig. 1 Identification of escape mutations by site-directed mutagenesis followed by selection and sequencing. a** The $bla_{ampC}$ gene of *E. coli* is mutated using MAGE technology to yield a mutant library of 790 different single amino-acid substitutions in the Bla$_{ampC}$ pocket. **b** The mutant library is then selected on a gradient of beta-lactam antibiotics with or without avibactam (AVI) and cell density is measured (Gray shade). At the antibiotic concentration that inhibits the growth of the WT bacteria ($MIC_{AB}^{WT}$, white cells), only resistant mutants grow and a decline in cell density is observed (light gray). When growth medium is supplemented with avibactam (+AVI) the growth of all mutants may be inhibited at drug concentrations lower than $MIC_{AB}^{WT}$ (middle row, left-pointing arrow) unless escape mutants are present, which grow on concentrations higher than $MIC_{AB}^{WT}$ even in the presence of avibactam (bottom row, right-pointing arrow). **c** Such escape mutations, as well as resistance-conferring mutations, are identified by high-throughput sequencing of the resulting culture.

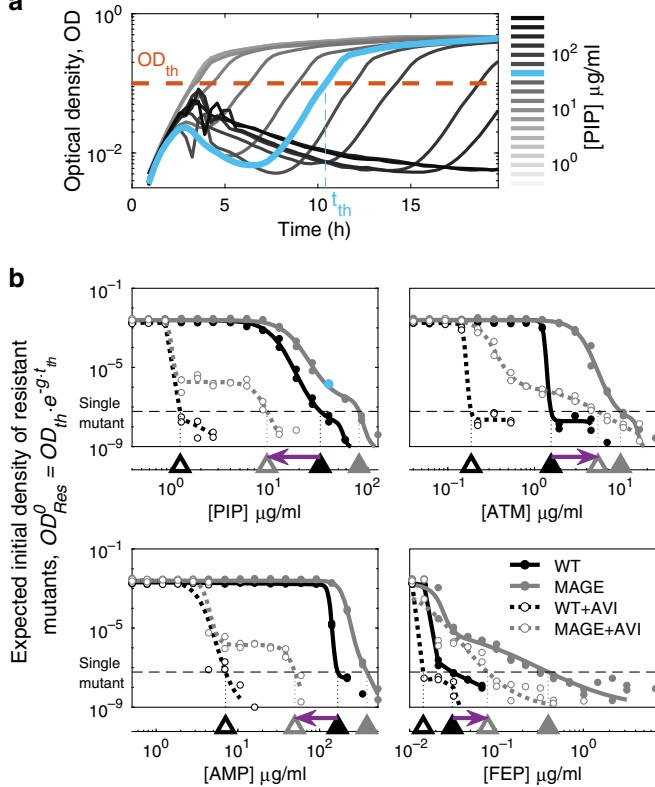

**Fig. 2 Combined with avibactam, different drugs differ substantially in the potential for escape mutations. a** Growth of the MAGE mutant library on a gradient of PIP concentrations plotted as Optical Density (*OD*) as a function of time. The time $t_{th}$ at which culture density crosses a set OD threshold=0.1 is determined ($OD_{th} = 0.1$). **b** The initial density of resistant mutants$OD_{Res}^0$ calculated based on $t_{th}$ and plotted for the WT (black) and the MAGE library (grey) on gradients of four beta-lactam antibiotics individually (filled symbols, 40 μg/ml PIP in cyan corresponds to panel **a**) and when supplemented with AVI (no-fill symbols). The drug concentration $\eta$ beyond which the value of $OD_{Res}^0$ drops below a density corresponding to a single initial mutant is determined (triangles on the bottom axis). The shift in the concentration required to inhibit the mutant library in the presence of AVI, compared to that required for the WT in the absence of AVI, is indicated by a purple arrow. A left-pointing arrow indicates that at the presence of AVI, no single mutant could grow at the drug-only MIC, while a right-pointing arrow is an indication for escape mutants. Source data are provided as a Source Data file.

of cells of a typical MAGE mutant ($OD_{Res}^0(\eta) = OD_{single\ mutant}$, where $OD_{single\ mutant}=3.3 \times 10^{-8}$, Methods). For each of the tested antibiotics (AB), we interpolated this critical drug concentration needed to prevent the growth of all initially present mutants for both the wild-type ($\eta_{AB}^{WT}$) and the MAGE library ($\eta_{AB}^{MAGE}$, Fig. 2b). We also subjected these same cultures to gradients of the same drugs in the presence of 0.25 μg/ml avibactam and determined the corresponding critical drug concentrations ($\eta_{AB}^{WT}|^{AVI}$, $\eta_{AB}^{MAGE}|^{AVI}$, Fig. 2b).

Wild-type Bla$_{ampC}$ mediated resistance varied substantially for different drugs, as well as its potential for resistance improving mutations. Supplementing cultures with avibactam lowered the resistance of the wild-type to piperacillin, aztreonam, cefepime and ampicillin (Fig. 2b), but not to meropenem (Supplementary

Fig. 4) indicating that active Bla$_{ampC}$ confers differential resistance to these drugs. Considering the phenotypes of the wild-type $bla_{ampC}$ and the MAGE mutant library without avibactam, we found that while no intragenic mutations in $bla_{ampC}$ led to increased ampicillin resistance, resistance to the other three drugs vastly improved (Fig. 2b, triangles). We next asked whether these mutations that increase resistance to piperacillin, aztreonam and cefepime in the absence of avibactam can possibly also escape the drug-inhibitor tradeoff.

Contrasting the drug concentration required to inhibit the mutant library in the presence of avibactam with the native inhibitory concentration, we find that escape mutations are accessible for some drugs and are not accessible in others. While no escape mutants are observed for piperacillin ($\eta_{PIP}^{MAGE}|^{AVI}<\eta_{PIP}^{WT}$, left-pointing arrow, Fig. 2b), escape mutations do appear for aztreonam and cefepime ($\eta_{ATM}^{MAGE}|^{AVI}>\eta_{ATM}^{WT}$ and $\eta_{FEP}^{MAGE}|^{AVI}>\eta_{FEP}^{WT}$, right-pointing arrow, Fig. 2b). To identify the genetic determinants of such escape mutations, we next analyzed the specific mutations

leading to resistance to beta-lactam drugs and to drug-inhibitor combinations.

**Drug-resistant mutants identified by amplicon sequencing.** To identify resistance-conferring mutations, we applied amplicon deep sequencing to a mutant library selected on gradients of beta-lactam drugs. First, to allow Illumina sequencing of the entire MAGE library, we pooled the 44-residue MAGE mutant libraries into two pools based on their position in the primary sequence. Second, we enriched these two pools for resistant mutants by selection on gradients of beta-lactam drugs with and without avibactam, then pooling cultures that grew on high drug concentrations to establish "enriched mutant libraries". Finally, we subjected duplicates of these two enriched mutant libraries on gradients of different beta-lactams with and without avibactam, recorded OD and chose a comprehensive subset of cultures for amplicon sequencing (Supplementary Fig. 5; Illumina MiSeq, Methods). Analyzing the amplicon reads, we identified, for each condition, the abundance of the WT and each of the mutants (Methods). Then, to estimate the resistance level of each mutant to each of the drugs ($IC50_{AB}^{Mut}$), we fitted the measured abundance data of each of the mutants across all drug concentrations with a dose-response model and determined the drug concentration that inhibits the mutant growth by 50% (Fig. 3a, Methods; Supplementary Fig. 6; Supplementary Data 2).

Analyzing the inhibitory concentrations of each of the mutants, we found that resistance to different drugs is mediated through only partially overlapping sets of mutations. Determining the IC50 of all mutants ($IC50_{AB}^{Mut}$), we identified, for each of the drugs, residues for which multiple beneficial substitutions exist (Fig. 3a, R148 for FEP; Y150 for ATM; E272 and N289 for PIP; S287 for both PIP and FEP). The diversity of beneficial substitutions for each of these residues indicates that these positions were maladapted (Fig. 3a). These residues were usually essential for Bla$_{ampC}$ activity on other drugs, possibly explaining their maladaptiveness (Fig. 3b). We further identified many mutations conferring high levels of resistance to both piperacillin and cefepime, but no cross high-resistance between aztreonam and the other drugs was found (Fig. 3b). However, a single mutation, E272I, had a small effect that extended over the three drugs (Fig. 3b). To link the mutations that affect drug resistance to the enzyme structure and its active site, we computationally docked each of our compounds to a free Bla$_{ampC}$ enzyme (AutoDock Vina[46]; Fig. 3c; Methods). We found that while substitutions of residues in loop H10 (Fig. 3c green, residues 287–296) to Glycine or Proline tend to confer resistance to both piperacillin and cefepime, they tend to have a negative effect on resistance to aztreonam (Fig. 3a). Such mutations are expected to destabilize the helix structure into a loop[47] and hence can make more space for piperacillin and cefepime that carry a large ring adjacent to the beta-lactam ring. Similarly, we found multiple substitutions of the conserved Y150 residue sensitizing towards cefepime and piperacillin, yet conferring resistance to aztreonam. These resistance-conferring mutations were next examined for their ability to escape the drug-inhibitor tradeoff.

**Escape mutations are rare and drug-specific.** Pairing our measurements of mutant inhibitory concentrations for drug-only and drug-inhibitor combinations, we characterized the drug-inhibitor resistance tradeoffs and identified escape mutations. For each mutant, we compared the changes in resistance to the drugs with and without avibactam relative to the wild-type (Fig. 4; Supplementary Data 2).

As expected, mutants with increased resistance to the beta-lactam drugs were often more sensitive to avibactam and mutants with increased resistance to avibactam became more sensitive to the beta-lactam drugs (Fig. 5a-d). Mutants with increased resistance to both the drug and the drug-avibactam combination also appeared, but commonly these mutants too were not able to grow at the native inhibitory concentration required to inhibit the wild-type in absence of avibactam (Fig. 5b-d; orange shade with blue horizontal lines in Fig. 5a). Yet, importantly, some rare mutations did confer resistance to the drug beyond the native inhibitory concentration of the wild-type even in the presence of the inhibitor (1.2% of the mutants presented $IC50_{AB}^{Mut}|^{AVI}>IC50_{AB}^{WT}$; red vertically-dashed area Fig. 5a; Fig. 5c, d). In agreement with our phenotypic measurements (Fig. 2b), these escape mutants were only identified for aztreonam and cefepime and not for piperacillin. Moreover, even for cefepime and aztreonam, such mutations are rare and drug-specific. While substitutions of the conserved Y150 to non-aromatic positively charged amino acids (R, K), or small amino acids (G and A), as well as to S and P can escape the tradeoff between resistance to aztreonam and avibactam, they are sensitizing to cefepime and do not confer resistance to the cefepime-avibactam combination (Fig. 5c, d; each such substitution appears in all accessible codons in all drug concentrations). Conversely, mutations that alter N346 to the large volume amino acids F, Y, W as well as the S237H and R148P substitutions can escape the tradeoff between resistance to avibactam and cefepime but are sensitizing to aztreonam and hence do not escape the avibactam-aztreonam combination (Fig. 5c, d). While all amino-acid substitutions in our library are derived from more than one nucleotide change and hence less accessible evolutionary, three of the escape mutations identified are also accessible by a single nucleotide change (Y150S for aztreonam as well as N346Y and R148P for cefepime).

To validate and better characterize the escape phenotype, we constructed strains expressing two two of these mutant alleles, Y150A and N346W, under an inducible promoter, and challenged them on a 2-D gradient of avibactam and either aztreonam or cefepime respectively (Methods). Identifying the IC50 isobole, the line in concentration space where bacterial growth is inhibited by 50% (Fig. 5e, f; Supplementary Fig. 7; Methods), we found that these escape mutations indeed increase the resistance to the inhibitor-drug combination even above the high levels needed to kill the wild-type without avibactam.

To further estimate the prevalence of escape mutations, we tested five additional beta-lactam antibiotics: the penicillins oxacillin (OXA) and penicillin G (PEN) as well as the cephalosporins cefoxitin (FOX), cefazolin (CFZ), and ceftazidime (CAZ). We inoculated the MAGE mutant pools on gradients of each of the five drugs with and without AVI, measured bacterial density over time and calculated $\eta$, the critical concentration that inhibit the growth of pre-existing mutants (Methods; Supplementary Fig. 8). While weak escape phenotype appeared for ceftazidime, all other tested beta-lactams prevented substantial resistance to the drug-inhibitor combinations (Fig. 5g). The mutants that escaped the ceftazidime-avibactam tradeoff (R148N and N346P; Methods) were similar to mutations that escaped the tradeoff between avibactam and cefepime, which is also a cephalosporin (R148P and N346F/Y/W), suggesting fine-tuning of resistance to the specific beta-lactam class. Overall, of the ten tested antibiotics escape mutants were found only for aztreonam, cefepime, and ceftazidime highlighting the rarity of such mutations.

**Discussion**

Systematically screening $bla_{ampC}$ beta-lactamase mutants, we identified mutations that escape the tradeoff between resistance to

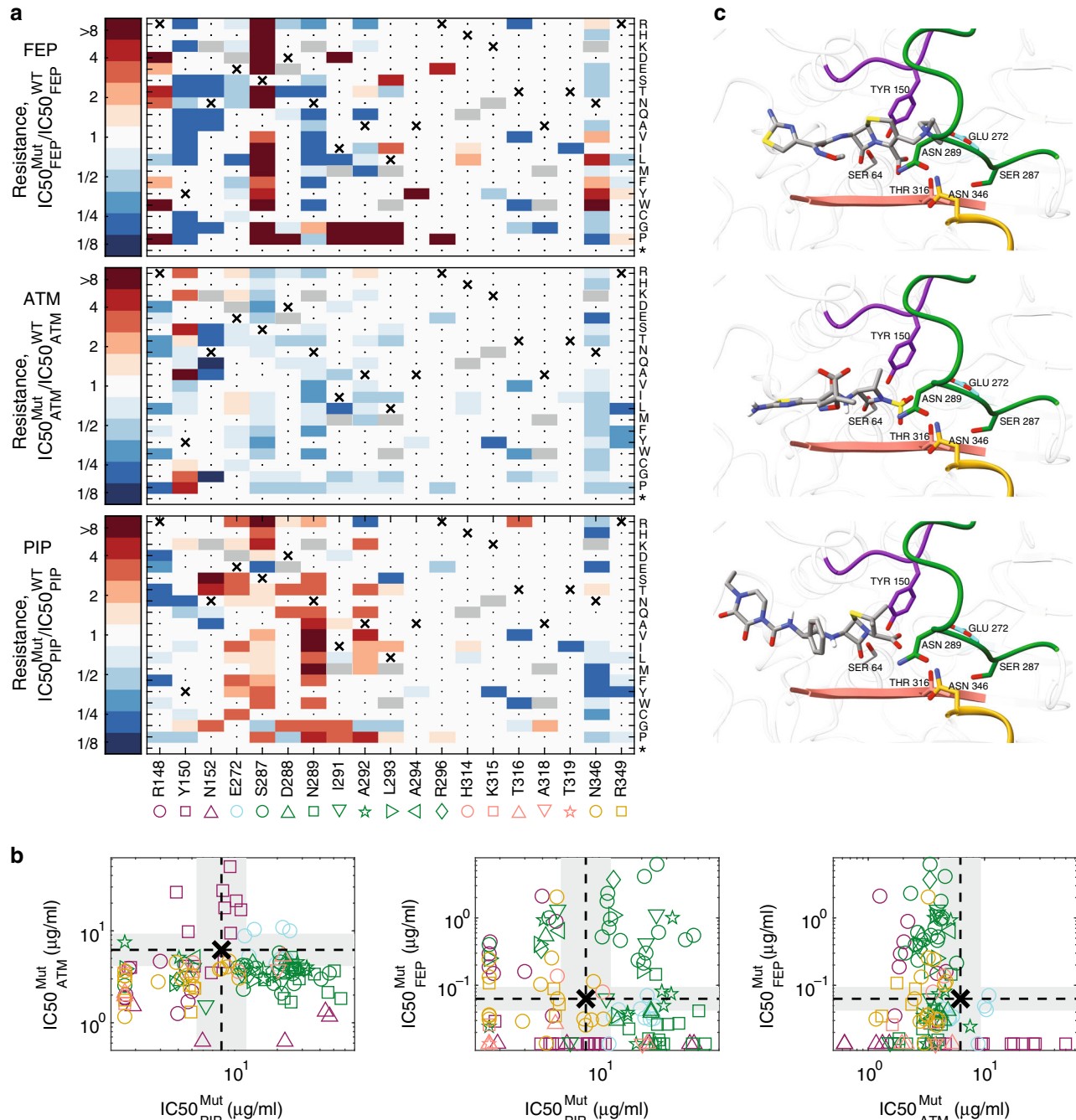

**Fig. 3 Mutations that confer resistance to one drug are often sensitizing to other drugs. a** For each of the 122 identified mutants we calculated resistance to FEP, AZT, and PIP as the drug concentration that inhibits mutant growth, $IC50_{AB}^{Mut}$, divided by that of the WT $IC50_{AB}^{WT}$. The WT amino acid is marked with an X and mutations, which were not included in our design are marked in gray. Mutations which did not confer resistance to any drug were not enriched and could not be identified in our data are marked with a dot. The resistance level of the latter to all drugs is expected to be similar to the WT or smaller. Source data are provided as a Source Data file. **b** The inhibitory concentration of each mutant is plotted on a logarithmic scale for the different drugs to assess cross-resistance. WT inhibitory concentration is marked with a dashed line and a gray background represents our resolution as determined by drug dilution factor between measurements. **c** The predicted conformation of the three antibiotics within the $Bla_{ampC}$ pocket. The antibiotics were docked using AutoDock Vina to the 3D crystal structure of $Bla_{ampC}$ beta-lactamase (PDB 1KVL) with the S64G replacement to emulate the WT physico-chemical environment. The enzyme is shown in ribbon, and regions that contain residues that affect resistance are colored in the same colors as in 3a-b. Residues for which multiple substitutions confer antibiotic resistance are highlighted and shown in stick representation. Antibiotics are colored-coded by atoms (gray for C, red for O, blue for N, yellow for S, and white for H).

beta-lactam drugs and to a beta-lactamase inhibitor. Our measurements show that such escape mutations appear only for some drug-inhibitor combinations, most prominently for FEP-AVI and ATM-AVI, and are rare and drug-specific. In these evolutionary susceptible drug-inhibitor combinations, even a single amino-acid change in the beta-lactamase enzyme $Bla_{ampC}$ can confer resistance levels exceeding the native concentration that the wild-type can sustain without the inhibitor. In contrast, we identified no single amino-acid mutants which escapes the piperacillin-avibactam tradeoff.

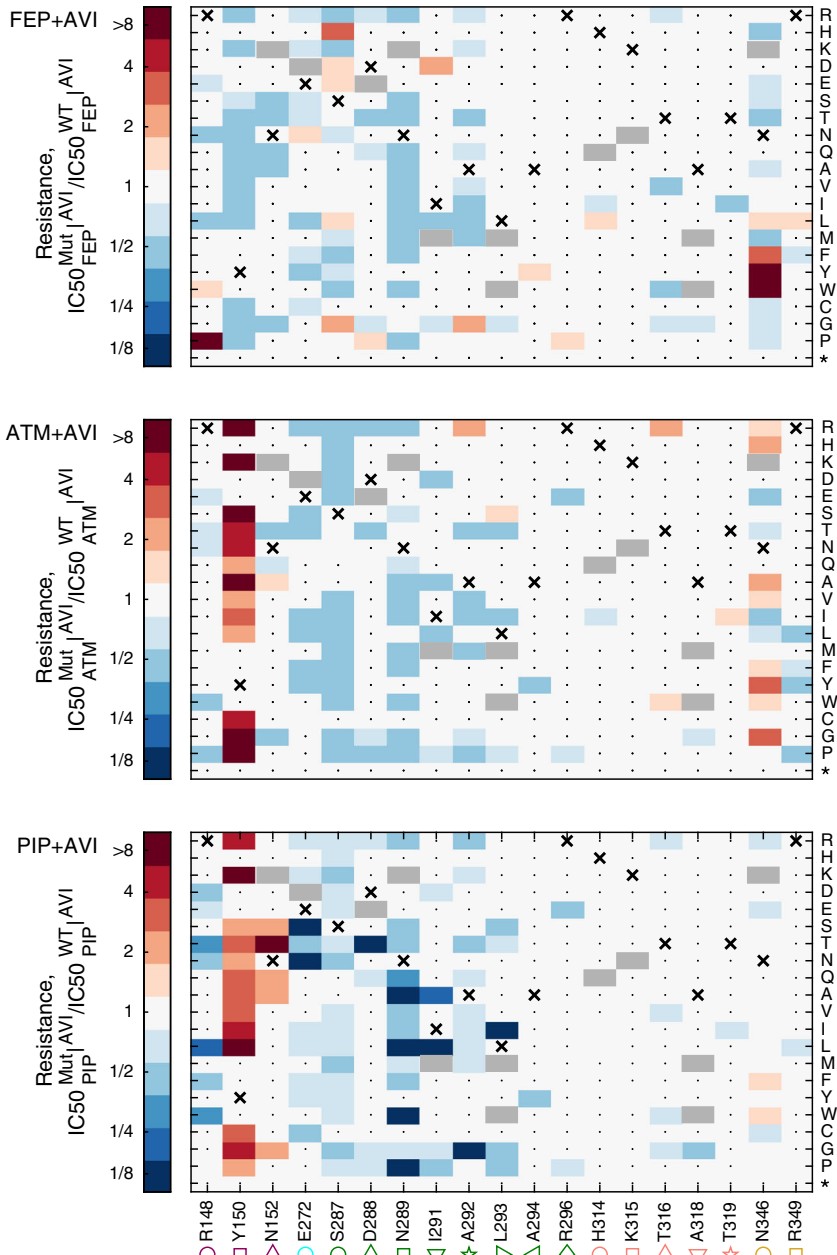

**Fig. 4 The resistance of all mutants to PIP, ATM, and FEP when co-applied with AVI as calculated from high-throughput sequencing data.** The relative resistance of all identified mutants to the beta-lactam drugs when supplemented with 0.25 μg/ml avibactam calculated as the drug concentration that inhibits mutant growth, $\mathrm{IC50}_{AB}^{Mut}|^{AVI}$, divided by that of the WT $\mathrm{IC50}_{AB}^{WT}|^{AVI}$. The WT amino acid is marked with an X and mutations, which were not included in our design are marked in gray. Mutations which did not confer resistance to any drug were not enriched and could not be identified in our data are marked with a dot. Source data are provided as a Source Data file.

The presence of escape mutations in specific drug combinations and their absence in others have implications for both mechanistic understanding and for refining diagnostics and treatment. The escape mutations identified were associated with the lack of a ring adjacent to the beta-lactam ring for aztreonam and with the stabilization of the ring in the R1 side-chain for cefepime. The lack of these two features in piperacillin may provide hints for the constraints of mutations that escape the piperacillin-avibactam tradeoff, yet specifically determining the structural mechanism remains a subject of future research. Independently of the mechanism, the specifically tested escape mutants enjoyed an ability to grow in the native inhibitory

concentration of the beta-lactam drug at a wide range of avibactam concentrations. These robust phenotypes suggest that escape mutations can potentially jeopardize the treatment efficacy of bacterial pathogens by drug-inhibitor combinations. Future studies are needed to determine for each beta-lactam drug which of the different beta-lactamase inhibitors form a treatment combination that is most resilient to evolutionary escape. Treatment optimization can also be based on genome-based diagnostics which use the escape mutations as predictive markers. While further investigations of escape mutations in vitro and in vivo are required, our identification of drug-inhibitor combinations permissive and non-permissive to escape mutations can

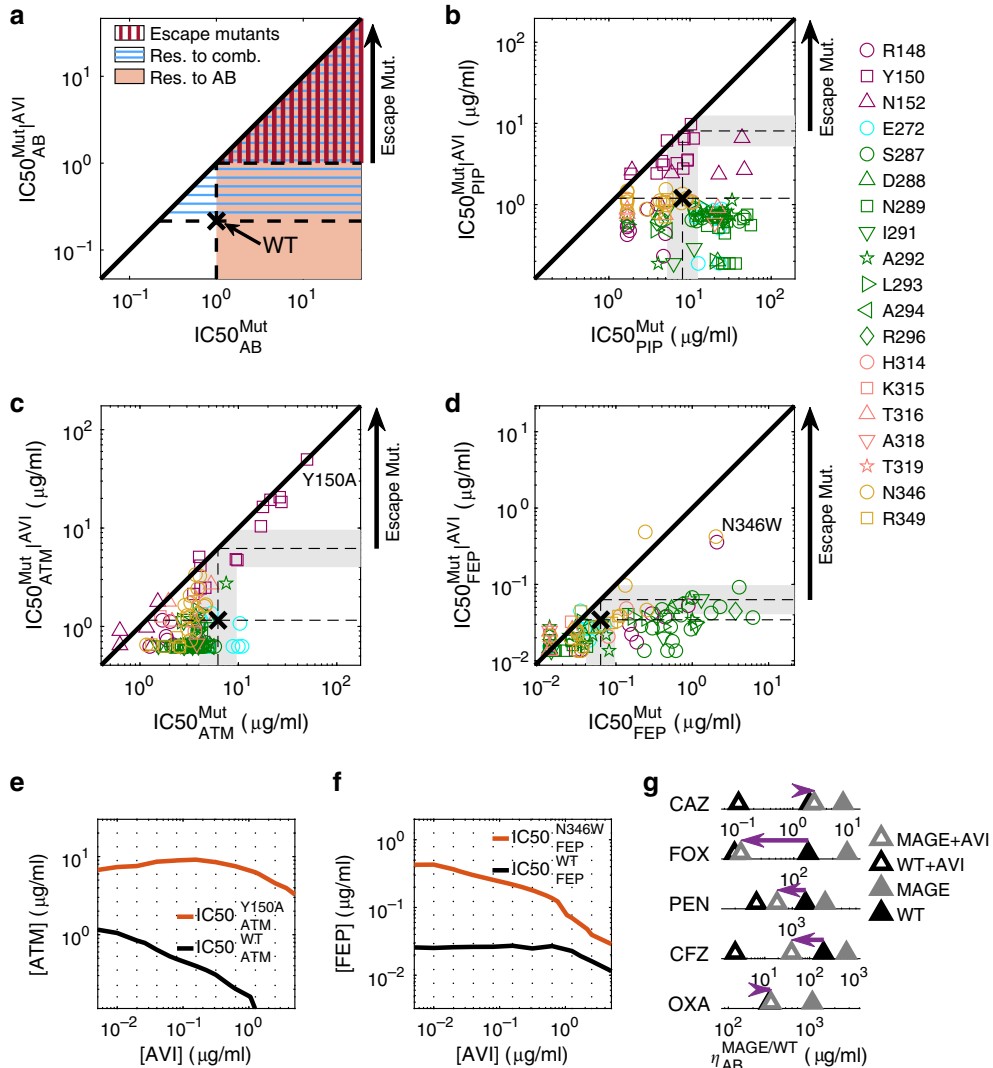

**Fig. 5 Escape mutations are rare and drug-specific. a** Areas in the concentration space of an antibiotic, AB, and antibiotic-AVI combination are highlighted by the resistance required to grow in them. Antibiotic-resistant mutations (orange shade, mutants are more resistant to the AB than the WT), combination-resistant mutations (blue horizontal lines, more resistant to AB than WT in the presence of AVI) and escape mutations (red vertical lines, more resistant to AB in presence of AVI than WT is to AB without AVI). **b–d** Mutants plotted according to their resistance to PIP (b), ATM (c), and FEP (d), when applied in isolation or supplemented with 0.25 μg/ml AVI. While no escape mutants appeared for PIP, a small subset of the mutants escaped the ATM-AVI or the FEP-AVI tradeoff. Those escape mutations were different between FEP and ATM. **e–f** The IC50 isoboles of a WT strain (black) and an escape mutant (red) for ATM (**e**) or FEP (**f**) are calculated based on 192 duplicate growth measurements (dots) and plotted on the 2D concentrations gradients of the drug and AVI. **g** The drug concentration required for inhibition of all pre-existing mutants, $\eta$, was measured for the WT strain as well as for the mutant library challenged by additional five drugs. Of those drugs, only ceftazidime (CAZ) present the weak escape phenotype. Source data are provided as a Source Data file.

serve as basis for an evolutionary-informed choice of combination drug treatments which minimize the potential for resistance.

## Methods

**Identifying the Bla$_{ampC}$ catalytic pocket.** Residues delineating the catalytic pocket of Bla$_{ampC}$ were selected by their distance from the ceftazidime drug bound to Bla$_{ampC}$ enzyme in the 3D crystal structure (PDB ID: 1IEL). We defined two layers according to their distance to the drug, residues within 5Å from the bound drug form the first layer as many of them serve as hotspot for substrate binding or catalytic activity, while more distant residues within a distance from 5Å to 7Å from the drug form the second layer, which are generally linked to the maintenance of the pocket 3D structure.

**Preparing a MAGE-compatible bla$_{ampC}$-expressing E. coli.** E. coli MG1655 bacteria were transformed with the pORTMAGE system[45], an inducible MAGE system transferable on a plasmid. Next, five bases in the proximal promoter (−32, −18, −1, +5, +58) of bla$_{ampC}$ were mutated by a MAGE protocol to activate bla$_{ampC}$ expression[48].

**Constructing the mutant library.** We aimed at altering each codon of interest into all amino acids accessible by a two or three nucleotide change from the original codon. To do so in a cost-effective way we used two oligos for each codon. Those 90 bases ssDNA oligos have flanking sequences identical to the genomic sequence, with three ambiguous bases to alter the codon of choice (Supplementary Data 1; oligos MS_AmpC_{position}_{i}_p{j} where {position} describes the mutated residue in the gene including the 16 amino-acid signaling peptide, {i} is 1 or 2, and {j} is the number of the MAGE pool. Positions within 45 bases of each other were transformed together as a pool since due to the proximity every mutant will carry only a single amino-acid change). In both oligos, the last of these ambiguous bases, aimed at the wobble position of the codon, is a mix of the three non-native nucleotides (if the native nucleotide is A, the oligo contains either C, G or T in a 1:1:1 ratio). One oligo presents a mix of the three non-native codons in the first position and a mix of all four nucleotides in the second position, while the other oligo presents the four nucleotides in the first position and three non-native ones in the second. For example, an AGC codon code for Serine is altered by oligos with BND and NHD mixes where N stands for any of the four nucleotides, B for C/G/T, D for A/G/T, and H for A/C/T. That way we get 45 codons out of the 54 possible codons that satisfy at least two nucleotide differences from the original codon, yet we lose only 4% of the possible amino-acid substitutions (Supplementary Fig. 2).

Each position of interest was targeted by two such oligos, and positions close enough to prevent double mutants were transformed together (Supplementary Data 1, pools p1-p10) by a standard MAGE protocol. In short, bacteria were grown in 2 ml LB at 30 °C to an optical density of 0.3 followed by activation of the MAGE system at 42 °C for 10 min. Next, the culture was put on ice for five minutes, washed twice in 50 ml double distilled water and transformed with ssDNA oligos by electrophoresis (1800V, 25μF, 200Ω) followed by recovery in rich medium (24 g/l Yeast extract, 12 g/l Tryptone, 9.4 g/l K$_2$HPO$_4$, 2 g/l KH$_2$PO$_4$). To enrich the mutant frequency, seven cycles of MAGE were applied before the libraries were pooled.

**Preparation of amplicon libraries and high-throughput sequencing.** The DNA from cultures chosen for sequencing was extracted (NucleoSpin Tissue, Manchery-Nagel) and amplicon libraries were prepared by two consecutive PCR reactions targeting the regions of sequencing pool 1 (sp1, 141-624 in the $bla_{ampC}$ gene) and sequencing pool 2 (sp2, 768-999 in the $bla_{ampC}$ gene). A first, short reaction of 10 cycles was conducted with primers that target the genomic sequence and add flanking sequences identical to the 5' region of Illumina adapters. Amplicon libraries have very low diversity, so to improve sequencing, diversity was added by designing primers with three phases from each direction (Supplementary Data 1, ampC_sp1_int F 1:3, ampC_sp1_int R 1:3, ampC_sp2_int R 1:3, ampC_sp2_int R 1:3). Next, the product was cleaned (AMPure XP beads, Beckman Coulter, by standard protocol) and was then used as a template for the second reaction of 25 cycles with primers that target the flanking ends of the product of the first PCR and contain Illumina adaptors and barcodes for sequencing (Supplementary Data 1, nx_i5, nx_i7). The products of the second reaction were cleaned and pooled equi-molarly (product concentration measured by Quant-iT, Thermo Fisher Scientific Q33130). Amplicon libraries were sequenced using Illumina MiSeq (250 paired-end using V2 chemistry) to yield a total of $7.5 \times 10^6$ reads. The sequencing reads covered the full amplicon of sequencing pool 2 and the majority of the amplicon of sequencing pool 1, except for a 50 bases gap in the middle. To cover this gap we additionally sequenced this library on Illumina MiSeq using custom intra-amplicon sequencing primers (Supplementary Data 1, ampC_seq_sp1 F, ampC_seq_sp1 R) using 150 bp nano protocol to yield smaller libraries with a total of $8.4 \times 10^5$ reads.

**Amplicon sequence analysis.** Data were analyzed using MATLAB custom scripts. The sequencing data were demultiplexed into samples with a mean coverage of 18,000 with standard deviation of 11,000 reads per library in the libraries of the first sequencing and a mean coverage of 2500 with a standard deviation of 1500 in the second. Paired reads from sequencing pool 2 were merged, primers of all samples were trimmed and reads were quality filtered (reads with more than 1% expected error were discarded). Next, the reads of a control strain that was added to the DNA extraction and carry the sequence ATGAGC at positions 236–241 and 917–922 were removed. In the cleaned data, the different mutants were identified when a read had a codon with two mutated bases. Reads that had no such codon represent a WT bacteria. The position of the mutated codon, as well as the sub-stituting amino-acid, were identified and the final relative abundance of each mutant in all conditions was calculated. Rare mutants covering less than $10^{-4}$ of the reads in the enriched mutant library are removed to avoid the Poisson noise of their inoculum.

**Calculating mutant resistance levels from abundance data.** To determine the IC50 of each mutant, we fitted the mutant abundance across the drug gradient to an expectation from a mathematical model accounting for the resistance level of each mutant and the competition among them. The model assumes a Hill-like dependence of the growth rate $g_i(c)$ of each mutant $i$ on drug concentration $c$, $g_i(c) = g^{max}/(1 + (c/IC50_i)^h)$, where $g^{max}$ is the growth rate in the absence of the drug, $IC50_i$ is the drug concentration that inhibits growth by 50%, and $h$ is the Hill coefficient of the drug. The abundance of each mutant at the end of growth is simply $N_i^{final}(c) = N_i^{initial}\exp(g_i(c) \cdot t^{final})$ where $N_i^{initial}$ is the initial abundance of the mutant as observed in the sequencing data of non-treated saturated cultures divided by the dilution factor been used to start the experiment (1:50), and $t^{final}$ is the duration of bacterial growth. We assume that the culture grows exponentially until the sampling time, $t^{sampling} = 16$ h, unless already before this time the culture density reaches a saturation level $N_{Total}^{Saturation}$. The total growth time is, therefore, $t^{final} = \min(t^{sampling}, t^{Saturation})$, where $t^{Saturation}$ is defined by $N_{Total}^{Saturation} = \sum_i N_i^{initial}\exp(g_i(c) \cdot t^{Saturation})$. The inhibitory concentrations for each drug $IC50_i$ were determined by fitting the model expected abundances $N_i^{final}(c)$ to the corresponding deep sequencing-based duplicate measurements of each mutant at each drug concentration. Specifically, we used nonlinear fitting to minimize the error of predicted growth of all mutants in all cultures. These fitted $IC50$ is used as a measure of drug resistance.

**Docking the beta-lactam drugs to a Bla$_{ampC}$ structure.** To generate putative binding poses, we docked antibiotics using the AutoDock Vina software package version 1.1.2 (May 11, 2011) with the default scoring function. In the AutoDock Vina configuration files, the parameter num_modes was set to 20 and

exhaustiveness to 48 to improve the searching space and accuracy. We identified the enzyme pocket based on the location of cephalothin on PDB ID 1KVL. We chose all the rotatable bonds in ligands to be flexible during the docking procedure, and we kept all the protein residues rigid. We assigned the Gasteiger atomic partial charges to the protein using the AutoDockTools package[49]. Antibiotics with their closed beta-lactam ring configuration were obtained from ZINC15 database[50] and modified by Maestro (Schrodinger Inc. Release 2019, LLC, New York, NY.2018) if necessary. Ligands and enzyme were converted to PDBQT format. From the docking results, we chose those poses with the best energy and a similar 3D configuration of the beta-lactam ring when compared to cephalothin from 1KVL (first for FEP, second for PIP and 11th for ATM in the energy rank) and further analyzed with ChimeraX[51].

**Measuring escape phenotype for Y150A and N346W mutants.** To validate the escape phenotype of $bla_{ampC}$ mutants Y150A and N346W, both mutants and a wild-type $bla_{ampC}$ control were cloned and measured directly. The open reading frame of mutant and wild-type genes were synthesized and cloned into pUC57-Kan (GENEWIZ). The open reading frames were then amplified. During PCR EcoRI and BamHI restriction sites as well as a ribosomal binding site were added. The genes were cloned into the multiple cloning site of the pSTV28 plasmid under an IPTG inducible lac promoter.

**Calculating the IC50 isobole from growth data.** Duplicate measurements of bacterial density after overnight growth in the presence of 30uM IPTG and under a range of drug concentrations, $D$, and avibactam concentrations, $I$, were taken, $OD(D,I)$. The IC50 isobole is the line in concentration space where bacterial growth is inhibited by 50%, $OD(D,I) = OD(0,0)/2$.

**Reporting summary.** Further information on research design is available in the Nature Research Reporting Summary linked to this article.

## Data availability
The source data underlying all Figures are provided as a Source Data file. The nucleotide sequence datasets generated during and/or analysed during the current study are available in the European Nucleotide Archive (ENA) repository under accession code PRJEB36781. Further growth measurements are available on the lab website (https://kishony.technion.ac.il/resources/).

## Code availability
Matlab scripts used in the current study are publicly available on the lab website (https://kishony.technion.ac.il/resources/).

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

## Acknowledgements

We thank Viktoria Lazar, Ákos Nyerges and Csaba Pal for bacterial strains and support with mutagenesis and Barak Shaer for graphic support. The work was supported in part by the US National Institutes of Health grant R01-GM081617 and by a grant from F. Hoffmann-La Roche Ltd (to R.K.).

## Author contributions

D.R., M.B., E.K., C.Z., A.H., and R.K. designed the project, D.R., E.S.T., and I.Y. conducted the experiments, D.R. and F.G. analyzed the data and D.R., I.Y., and R.K. wrote the manuscript with feedback from all authors.

## Competing interests

The authors declare no competing interests.
