## [Peer Review File · Nature Communications]

Reviewers' comments:

Reviewer #1 (Remarks to the Author):

The evolution of antimicrobial resistance is an issue of global concern. Beta-lactamases are exposed to the evolutionary pressure of the indiscriminate use of antibiotics. Many studies of the clinical and in vitro evolution of lactamases have addressed this issue.

In this manuscript, Russ and coworkers report a mutational study of the class C lactamase AmpC exposed to the combined action of the beta-lactamase inhibitor avibactam, and different antibiotics. They find that escape mutations for these combinations are rare and drug-specific. From the fundamental point of view, this work contributes to the understanding of protein evolution, and also is of great relevance in setting the bases for possible treatments in the clinic. Overall, is an impressive amount of high quality work with sound data and conclusions.

There are several aspects that the authors may want to address to further improve the quality of this work:

1. One issue of concern is that codons are mutated by 2-3 nucleotides. This is rather unusual in natural evolution. Despite this approach does not disprove the findings of this work, the authors should comment on the possible bias on the results.

2. Surprisingly, no escape mutations were observed for piperacillin-avibactam. A structural or mechanistic-based hypothesis for this finding would enrich the discussion.

Reviewer #2 (Remarks to the Author):

In the manuscript "Escape mutations circumvent a tradeoff between resistance to a beta-lactam and a beta-lactamase inhibitor" the authors use site-directed mutagenesis to identify mutations that allow *Escherichia coli* to evade a trade-off between different beta-lactam drugs and avibactam, a beta-lactamase inhibitor. The high-throughput approach used by the authors is sound and the conclusions appropriately supported by the data.

In particular, the authors use Multiplexed Automatic Genetic Engineering (MAGE) to generate a comprehensive library of single amino acid substitutions in a chromosomal gene (*ampC*) conferring resistance to a wide range of beta-lactam antibiotics. The authors then expose the pooled library to avibactam in combination with a range of drug concentrations to identify the mutation-antibiotic pair where this trade-off is circumvented. Interestingly, the authors found mutants that were able to evade the trade-off in some cases (PIP, ATM, AMP, CAZ) but not in others (MER, OXA, PEN, FOX, CFZ), suggesting that escape mutations can be very drug-specific. These mutants were identified through amplicon deep sequencing after growing in a range of drug concentrations, and their drug susceptibility to the drug-inhibitor combination was obtained by comparing the IC₅₀ of the mutant with respect to that of the parental strain. While most of these mutants presented a lower IC₅₀ in the presence of avibactam, a few of them were more resistant in the presence of avibactam than even the parental strain without the beta-lactamase inhibitor.

Although there are some limitations to their study (for instance using a single avibactam concentration throughout the study), I believe this is an interesting manuscript that would be of interest to a large community of researchers and medical practitioners and, therefore, I recommend this paper for publication. I only have a few comments that I believe should be addressed prior to publication:

In Figure 1 the authors define drug-susceptibility based on the drug concentration whereby bacterial density is below the observable limit (a quantity referred to in the literature as the minimum inhibitory concentration, MIC). Later in Figure 2 they use a non-standard assay to determine drug-susceptibility based on the time elapsed before bacterial density reaches an arbitrary threshold. This measurement assumes that all mutants have the same growth rate as the

parental strain, an observation that seems to be consistent with the data. However, it would be interesting to evaluate the robustness of this measure to variations in this threshold. Later, in Figure 3, the authors quantify drug susceptibility based on the concentration where growth is half of the achieved by the population in the absence of antibiotics (IC₅₀, while in Figure 4 data is presented in terms of log(IC₅₀). I believe the results presented in this study hold independently of the definition used and, for this reason, I would recommend the authors to use a consistent measure of susceptibility throughout the manuscript.

The data presented in this study is exhaustive (a dozen drugs with and without avibactam probed in hundreds of mutants). However, as far as I can see, the authors only obtain a single replicate per experimental data point. This is concerning given that some of the experimental protocols used (notably dose-response experiments) are known to present huge experimental variability. This lack of statistical power contrasts with the use of subjective terms as "vastly", "rare", etc. I would recommend the authors to avoid using these ambiguous words and to use statistical tests applied to multiple biological replicates to support the significance of their results.

Finally, in the last paragraph, the authors argue that this study could be used as "a basis for evolutionary-informed choice of combination treatments that minimize the potential for resistance". This is a thought-provoking statement that may or may not be true but, in any case, I think the authors should further discuss what they mean by this or remove this statement completely to avoid sounding unnecessarily speculative.

Reviewer #3 (Remarks to the Author):

In this manuscript, the authors use the *E. coli* blaampC as a model system to study the evolution of resistance to beta-lactam-beta-lactamase inhibitor combinations. Typically, resistance to a substrate results in an evolutionary tradeoff with increased susceptibility to the inhibitor. The goal was to identify mutations that bypassed this phenomenon. This is highly relevant as in clinic these types of mutants have emerged to ceftazidime-avibactam. This wasn't predicted, historically resistance to BL-BLI combinations was the result of mutations leading to the BLI no longer inhibiting the enzyme. However, once ceftazidime-avibactam reached clinical use, a different story emerged. The enzymes evolved to be more resistant to the beta-lactam partner and even though the inhibitor still could inhibit the enzyme the combo failed. A similar situation is occurring with ceftolozane-tazobactam, but one may say this was more predictable as the cefotolozane was engineered to bypass the beta-lactamase that is already not susceptible to inhibition by tazobactam. The strategy used by authors to "choose" a beta-lactam partner for a beta-lactamase inhibitor is novel and of great interest to the field. Given the for profit nature of antibiotic development, it is unclear if approaches like this would ever be adapted as BLIs are sometimes paired with BLs for not the best scientific reasons.

Suggestions:

1. The authors may want to expand their discussion and include the relevance of their study to drug development.
2. I suggest including Supplemental Figure 7 in the main body of the manuscript.
3. Do the authors have any hypotheses about why some beta-lactams drive the escape phenotype and other do not? Or they can reference from the literature.
4. How did the authors control for any mutations that may have arisen outside of the blaampC gene as antibiotic pressure was present in the assay?

5. It would be helpful to report values in terms of MICs for those in the field. A table with the MICs should be added. What was the wildtype MIC and the mutants? This will help the audience understand the impact of these mutations.
6. There is a random blue dot on Figure 2b.
7. Please refer to beta-lactamase genes with the *bla* designation. *bla* is italicized and the gene *ampC* is in subscript.

Reviewers' comments:

Reviewer #1 (Remarks to the Author):

The evolution of antimicrobial resistance is an issue of global concern. Beta-lactamases are exposed to the evolutionary pressure of the indiscriminate use of antibiotics. Many studies of the clinical and in vitro evolution of lactamases have addressed this issue.

In this manuscript, Russ and coworkers report a mutational study of the class C lactamase AmpC exposed to the combined action of the beta-lactamase inhibitor avibactam, and different antibiotics. They find that escape mutations for these combinations are rare and drug-specific. From the fundamental point of view, this work contributes to the understanding of protein evolution, and also is of great relevance in setting the bases for possible treatments in the clinic. Overall, is an impressive amount of high quality work with sound data and conclusions.

We appreciate the reviewer's thorough reading and his support of the manuscript.

There are several aspects that the authors may want to address to further improve the quality of this work:

1. One issue of concern is that codons are mutated by 2-3 nucleotides. This is rather unusual in natural evolution. Despite this approach does not disprove the findings of this work, the authors should comment on the possible bias on the results.

Following the concern raised by the reviewer, we have now revised the results (4th section entitled "escape mutations are rare and drug-specific") and the discussion to address possible biases stem from avoiding single SNP mutations. We also underscore that our library was designed to cover with two-point mutations all possible amino acid substitutions (including those that could stem from single-nucleotide change), and that we have validated experimentally that we cover over 90% of all the possible amino-acid substitutions

2. Surprisingly, no escape mutations were observed for piperacillin-avibactam. A structural or mechanistic-based hypothesis for this finding would enrich the discussion.

This is indeed a very interesting point. While we feel that the specific mechanism is beyond the scope of this work, following the reviewer comments we have now added a couples of sentences in the conclusion pointing to the likely mechanism.

Reviewer #2 (Remarks to the Author):

In the manuscript "Escape mutations circumvent a tradeoff between resistance to a beta-lactam and a beta-lactamase inhibitor" the authors use site-directed mutagenesis to identify mutations that allow *Escherichia coli* to evade a trade-off between different beta-lactam drugs and avibactam, a beta-lactamase inhibitor. The high-throughput approach used by the authors is sound and the conclusions appropriately supported by the data.

We thank the reviewer's appreciation of the rigor of the methodology and the results.

In particular, the authors use Multiplexed Automatic Genetic Engineering (MAGE) to generate a comprehensive library of single amino acid substitutions in a chromosomal gene (*ampC*) conferring resistance to a wide range of beta-lactam antibiotics. The authors then expose the pooled library to avibactam in combination with a range of drug concentrations to identify the mutation-antibiotic pair where this trade-off is circumvented. Interestingly, the authors found mutants that were able to evade the trade-off in some cases (PIP, ATM, AMP, CAZ) but not in others (MER, OXA, PEN, FOX, CFZ), suggesting that escape mutations can be very drug-specific.

Following the reviewer comment, we noticed that the text was somewhat confusing. Specifically, escape mutations were found for ATM, CAZ and CFZ while for PIP only antibiotic resistance mutations were found. We have now reworded the text to make sure this is correctly stated (see revised text in the last paragraph of the Results section).

These mutants were identified through amplicon deep sequencing after growing in a range of drug concentrations, and their drug susceptibility to the drug-inhibitor combination was obtained by comparing the IC₅₀ of the mutant with respect to that of the parental strain. While most of these mutants presented a lower IC₅₀ in the presence of avibactam, a few of them were more resistant in the presence of avibactam than even the parental strain without the beta-lactamase inhibitor.

Although there are some limitations to their study (for instance using a single avibactam concentration through the study), I believe this is an interesting manuscript that would be of interest to a large community of researchers and medical practitioners and, therefore, I recommend this paper for publication. I only have a few comments that I believe should be addressed prior to publication:

We appreciate the reviewer's thorough reading and for his support of the manuscript.

We have now added **new experiments** demonstrating the escape mutant phenotype on complete 2-D gradients of avibactam and the beta-lactam antibiotics. Our measurement on this 2D gradient supports the escape phenotypes identified en masse on a single AVI concentration. We have now edited the results section and added a new experiment (**new Figure 5e,f and supplementary fig. 7**) to highlight that point.

In Figure 1 the authors define drug-susceptibility based on the drug concentration whereby bacterial density is below the observable limit (a quantity referred to in the literature as the minimum inhibitory concentration, MIC).

Later in Figure 2 they use a non-standard assay to determine drug-susceptibility based on the time elapsed before bacterial density reaches an arbitrary threshold. This measurement assumes that all mutants have the same growth rate as the parental strain, an observation that seems to be consistent with the data. However, it would be interesting to evaluate the robustness of this measure to variations in this threshold.

Following the reviewer's concern, we now added a sensitivity assay to validate that our measurements are independent of the threshold chosen (supplementary fig. 3b). We also provide the results of a new 2-D gradient experiments where the MIC is determined by the more direct and standard method based on raw OD measurements (**new experiment; Fig. 5e,f and supplementary fig. 7**).

Later, in Figure 3, the authors quantify drug susceptibility based on the concentration where growth is half of the achieved by the population in the absence of antibiotics (IC₅₀, while in Figure 4 data is presented in terms of log(IC₅₀). I believe the results presented in this study hold independently of the definition used and, for this reason, I would recommend the authors to use a consistent measure of susceptibility throughout the manuscript.

Following the reviewer's comment, we now have edited Figure 5a (old Fig. 4a) to be more consistent with the rest of Figure 5.

The data presented in this study is exhaustive (a dozen drugs with and without avibactam probed in hundreds of mutants). However, as far as I can see, the authors only obtain a single replicate per experimental data point. This is concerning given that some of the experimental protocols used (notably dose-response experiments) are known to present huge experimental variability. This lack of statistical power contrasts with the use of subjective terms as "vastly", "rare", etc. I would recommend the authors to avoid using these ambiguous words and to use statistical tests applied to multiple biological replicates to support the significance of their results.

We failed to explain that our data is based on duplicates. In the revised ms we have changed the wording of the Results and the Methods sections to reflect it and the effect size.

Finally, in the last paragraph, the authors argue that this study could be used as "a basis for evolutionary-informed choice of combination treatments that minimize the potential for resistance". This is a thought-provoking statement that may or may not be true but, in any case, I think the authors should further discuss what they mean by this or remove this statement completely to avoid sounding unnecessarily speculative.

Following the reviewer's comment, we supplemented the discussion section to discuss the importance of escape mutations to informed choice of antibiotic treatment.

Reviewer #3 (Remarks to the Author):

In this manuscript, the authors use the E. coli bla_{ampC} as a model system to study the evolution of resistance to beta-lactam-beta-lactamase inhibitor combinations. Typically, resistance to a substrate results in an evolutionary tradeoff with increased susceptibility to the inhibitor. The goal was to identify mutations that bypassed this phenomenon. This is highly relevant as in clinic these types of mutants have emerged to ceftazidime-avibactam. This wasn't predicted, historically resistance to BL-BLI combinations was the result of mutations leading to the BLI no longer inhibiting the enzyme. However, once ceftazidime-avibactam reached clinical use, a different story emerged. The enzymes evolved to be more resistant to the beta-lactam partner and even though the inhibitor still could inhibit the enzyme the combo failed. A similar situation is occurring with ceftolozane-tazobactam, but one may say this was more predictable as the ceftolozane was engineered to bypass the beta-lactamase that is already not susceptible to inhibition by tazobactam. The strategy used by authors to "choose" a beta-lactam partner for a beta-lactamase inhibitor is novel and of great interest to the field. Given the for profit nature of antibiotic development, it is unclear if approaches like this would ever be adapted as BLIs are sometimes paired with BLs for not the best scientific reasons.

We appreciate the reviewer's thorough reading and support of the manuscript.

Suggestions:

1. The authors may want to expand their discussion and include the relevance of their study to drug development.

Following the reviewer's comment, we now discuss the impact of the identified escape mutations on drug development in the discussion section.

2. I suggest including Supplemental Figure 7 in the main body of the manuscript.

Done.

3. Do the authors have any hypotheses about why some beta-lactams drive the escape phenotype and other do not? Or they can reference from the literature.

The mechanism is beyond the scope of our current MS but following the reviewer comments we have now added a couple of sentences in the conclusion pointing to the likely mechanism.

4. How did the authors control for any mutations that may have arisen outside of the bla_{ampC} gene as antibiotic pressure was present in the assay?

To address this very important concern, we conducted a new experiment with newly designed strains (**new experiment, Fig. 5e,f**). We constructed three new plasmids to express the WT Bla_{ampC} enzyme or one of the escape mutants Y150A or N346W under the inducible lac promoter and expressed them on a deleted endogenous bla_{ampC} gene background. We next measured the strains growth on 2D gradient of betalactam and avibactam. These experiments

confirm that the escape phenotype observed stems from the specific ampC mutations rather than from any off-target mutations.

In addition, we now note that the escape phenotypes we observed appear in multiple codons for most amino acid changes, demonstrating that independent clones show the same escape phenotype, suggesting that it does not stem from off-target mutations.

5. It would be helpful to report values in terms of MICs for those in the field. A table with the MICs should be added. What was the wildtype MIC and the mutants? This will help the audience understand the impact of these mutations.

Following the reviewer comment, we now refer to the inhibitory concentration values in supplementary table 2.

6. There is a random blue dot on Figure 2b.

That blue dot presents the expected density of viable mutants in the 40 ug/ml PIP treatment (corresponding to the cyan line in Fig. 2a). Following the reviewer's comment, we now note it in the figure caption.

7. Please refer to beta-lactamase genes with the bla designation. bla is italicized and the gene ampC is in subscript.

Done.

REVIEWERS' COMMENTS:

Reviewer #2 (Remarks to the Author):

The revised version of the manuscript is more precise and has addressed most of the comments by the reviewers.

Reviewer #3 (Remarks to the Author):

The authors have adequately addressed my concerns in this revision.

Response to referees:

Reviewer #2 (Remarks to the Author):

The revised version of the manuscript is more precise and has addressed most of the comments by the reviewers.

Reviewer #3 (Remarks to the Author):

The authors have adequately addressed my concerns in this revision.

We highly appreciate the constructive comments, help and support of the referees.